# Biomarkers for Lifetime Caries-Free Status

**DOI:** 10.3390/jpm11010023

**Published:** 2020-12-30

**Authors:** Ariana M. Kelly, Mariana Bezamat, Adriana Modesto, Alexandre R. Vieira

**Affiliations:** School of Dental Medicine, University of Pittsburgh, Pittsburgh, PA 15260, USA; amk211@pitt.edu (A.M.K.); mbl29@pitt.edu (M.B.); ams208@pitt.edu (A.M.)

**Keywords:** dental caries, edentulism, genomics

## Abstract

The purpose of this study was to address the hypothesis that extreme outcomes of dental caries, such as edentulism
or prematurely losing permanent teeth are associated with genetic variation in enamel-formation genes. After scanning 6206 individuals, samples of 330 were selected for this study. Tested phenotypes included patients who were edentulous by age 30, patients with missing first molars by age 30, patients with missing second molars by age 30, and caries-free patients. Fourteen single nucleotide polymorphisms were genotyped by TaqMan chemistry. The analyses of each phenotype were performed using the software PLINK with an alpha of 0.05. Nominal associations were found between rs12640848 in enamelin (*p* = 0.05), rs1784418 in matrix metallopeptidase 20 (*p* = 0.02), and rs5997096 in the tuftelin interacting protein 11 and being caries-free at the age of 60. When combining patients that were missing both first mandibular molars and missing both second mandibular molars, no associations were found. Matrix metallopeptidase 20, and tuftelin interacting protein 11 also showed trends for association with being caries-free. Genetic variation in *TFIP11*, *MMP20*, and *ENAM* may have a protective effect increasing the chances of individuals preserving their teeth caries-free over a lifetime.

## 1. Introduction

Dental caries is second only to influenza in the number of people it affects around the globe [1]. For more than 100 years, we have defined how its pathogenesis works. However, that knowledge did not allow for a total eradication of this highly preventable disease. Interventions for dental caries, aside from fluoridating the drinking water or application of sealants in school children, have been limited. Furthermore, the practice of fluoridating the drinking water will soon be challenged since there is evidence that prenatal exposure may impact cognitive performance [2].

The current challenge with dental caries is identifying individuals that are at higher risk of the disease. These individuals, who are approximately 25% of the total number of individuals affected, have 80% of the disease burden [3]. Since it is unrealistic to expect that interventions that can impact sugar consumption, such as taxation of sugar products, will be implemented relatively soon [4], and dentists are not trained to effectively implement behavioral-change interventions [5], the identification of markers that help identify individuals at higher risk are needed.

The focus on the host is potentially a strategy that may benefit the fight against dental caries. There is unquestionable evidence that dental caries has a genetic component, and genes associated with dental caries include the ones involved in mineralization of bone, formation of enamel, microbial colonization, and the degradation of amelogenin [6,7,8]. These studies so far included the typical dental patients that are found to have carious lesions, and the definition of the phenotype range from unsophisticated analyses of individuals with no lesions versus any number of lesions [9], self-reported tooth loss [10], individuals with no lesions versus ones with a substantial number of lesions [11], to a more sophisticated framework taking into consideration the number of lesions and the age of the individuals [12]. Likewise, our recent study demonstrated a potential genetic component present in cancer-diagnosed individuals who had tooth loss/edentulism [13]. However, the main limitation experienced was the lack of phenotype differentiation between losing one tooth or being edentulous. Here we hypothesized that creating definitions that clearly select individuals more severely affected will provide us with a chance to identify genetic markers that may serve as risk discriminators. Therefore, the purpose of this study was to test extreme outcomes of dental caries, such as edentulism at young age or prematurely losing permanent mandibular molars, to determine genetic biomarkers that can be used at the population level to identify individuals at higher risk for dental caries.

## 2. Materials and Methods

Beginning in September 2006, every individual that comes to the University of Pittsburgh School of Dental Medicine for treatment has been given the opportunity to be a part of the Dental Registry and DNA Repository project (University of Pittsburgh Institutional Review Board (IRB) approval # 0606091). This study conforms to the STROBE (Strengthening the Reporting of Observational Studies in Epidemiology) guidelines [14]. At the time of this analysis, there were 6206 subjects in the University of Pittsburgh School of Dental Medicine Dental Registry and DNA Repository project [5,15]. All individuals who agreed to participate gave written, informed consent authorizing the use of information from their dental and medical records. From the total of 6206 individuals participating in the registry, the records of 330 were obtained to perform the present study. The tested phenotypes included edentulous patients by age 30, patients with missing first molars by age 30, patients with missing second molars by age 30, and caries-free patients (Table 1). Missing teeth in all groups were due to dental caries. Individuals with missing teeth due to reasons other than dental caries were not included in the study.

Genomic DNA was extracted from salivary samples of the 330 individuals for the genetic analyses proposed for this study. Fourteen single nucleotide polymorphisms (*ESRRB* rs10132091 and rs6574293, *Defensin* rs11362, *ENAM* rs12640848, *MMP20* rs1784418, *KLK4* rs198968 and rs2235091, *ALOX15* rs2619112 and rs7217186, *PART1* rs27565, *AQP5* rs3736309, *Tuftelin 1* rs3790506, and *AMBN* rs4694075 and *SRRD/TFIP11* rs5997096) previously associated with dental caries in our studies [6,11,16,17,18,19,20,21,22,23,24,25] were genotyped using TaqMan chemistry (Applied Biosystems) [26]. The analyses of each phenotype, using sex and ethnicity as covariates were performed using the software PLINK (Center for Human Genetic Research (CHGR), Massachusetts General Hospital (MGH), and the Broad Institute of Harvard and MIT) [27]. The software generated the frequency of a particular variant in cases versus controls. An alpha of 0.05 was considered for nominal results and a Bonferroni correction was applied for multiple testing and *p*-values below 0.003 (0.05/14) were considered significant. This more stringent value aimed to retain the prescribed family-wise error rate alpha. The caries-free individuals served as reference (Table 2).

## 3. Results

Genotyping distributions of *TFIP11* rs5997096 in individuals who lost their second mandibular molars by age 30 was different from those of individuals who were caries-free by age 60 (*p* = 0.03), with caries-free individuals more likely to have one or two copies of the less common allele.

In the instances where genotyping distribution *p*-values were between 0.051 and 0.15, we tested for dominant and recessive models. Nominal associations were found between *ENAM* rs12640848 and edentulism in the dominant model (*p* = 0.05), as well as *MMP20* rs1784418 and edentulism in the recessive model (*p* = 0.02). Individuals caries-free by age 60 were more likely to have the less common allele of *ENAM* rs12640848 and two copies of the less common allele of *MMP20* rs1784418. Considering just the loss of one permanent mandibular molar (first or second, left or right) did not change these results. The differences in sex and ethnicity did not appear to influence the results.

## 4. Discussion

This report is the first explicit attempt to define genomic biomarkers for dental caries, approaching the problem by testing individuals with extreme scenarios, such as having lost all their teeth by age 30 or lost the 6- and 12-year mandibular molars by age 30, in contrast to a lifetime being caries-free. Looking at extremes of phenotypic presentations has been successfully used to study obesity [28], erosive tooth wear ex vivo [29], and dental caries experience [11,30]. We identified one genetic variant that has the potential to discriminate individuals who are at the lowest risk for losing teeth early in life and therefore remain caries-free for a lifetime, despite all odds.

The study of dental caries in humans has relied on phenotypical definitions that in many instances do not inform disease mechanisms but caries experience. Different from other conditions with higher heritability, the genetic variance that explains the burden of dental caries in populations is likely easily overcome by environmental factors. Therefore, comparing individuals who are caries-free with individuals with any number of lesions may limit the ability to identify biological components contributing to caries. Phenotypical definitions that rely on trajectories of disease [31] might be a more promising way to study the problem.

*MMP20* is a matrix metalloproteinase that modifies amelogenin, the primary enamel matrix secreted protein during ameloblast maturation [32,33]. We and others [23,34,35] showed earlier that the less-common allele of *MMP20* rs1784418 appears to have a protective effect against caries experience and these results are supported by our most recent findings in extreme phenotypes.

Another interesting finding is the association of *TFIP11* with being a lifetime caries-free individual. We have shown associations between *TFIP11* and formation of artificial caries subclinical lesions and speculated that this may be due to the influence of *TFIP11* in the dental enamel ability to reuptake fluoride from the environment [6,8]. This early finding is compatible to our results suggesting that *TFIP11* is associated with being caries-free.

Finally, *ENAM* rs12640848 was also associated with being caries-free over a lifetime. We previously showed that *ENAM* rs3796704 was associated with having a dental enamel that was softer [8]. Others have also shown missense mutations in *ENAM* linked to higher caries experience [36]. These data suggest that depending on the variant, enamelin can be either protective or can increase the susceptibility to dental caries.

Pittsburgh is the largest city adjacent to one of the poorest areas in the USA, Appalachia. The Appalachian mountain range extends across 13 states in the United States from New York to Mississippi. Socio-economic indicators are much worse for the communities in the Appalachian region compared to those in the rest of the United States [37]. In regard to health indicators, Pittsburgh reflects what is found in the Appalachian region, and the population treated at the University of Pittsburgh Medical Center has some of the worst health indicators in the country, which makes Pittsburgh a perfect laboratory for studying disease risks [38]. This may be particularly true for the study of dental caries. Since the disease has a heritability likely not much higher than 20%, factors such as diet, type of bacterial colonization, and oral hygiene habits are very relevant. However, in our cohort, socioeconomics of the patients was not dramatically different, and quality of oral hygiene and diet may be less variable in a group that averages more than 15 affected teeth [38]. On the other hand, the distinct profile of the population surrounding Pittsburgh may be too unique to make our findings generalizable and it is desirable to repeat these experiments in other parts of the world.

Being concerned about multiple testing, we applied the strict Bonferroni correction and none of the associations reached the determined threshold of *p*-values below 0.003. We have demonstrated before [39] that known true associations are missed when correction for multiple testing is implemented. The results of our work should be considered with caution and serve to generate hypotheses to be directly tested in larger and more homogeneous samples. At the same time, nominal associations found here should be considered as trends for associations and should not be ignored. Other perceived limitations of our study are the distribution of Blacks and Whites in some of the comparisons that may have been unequal, the relatively small group (*n* = 30) of edentulous individuals by age 30, and the absence of a direct measure of oral hygiene and diet.

Strategies for identifying individuals at higher risk for dental caries are still needed. Clinical findings such as Class II malocclusion, open bite, and dental crowding showed trends for associating with lower caries experience in the primary dentition [40]. Genetic associations may only be unveiled under a common dysbiotic scenario following local or systemic changes [41,42,43], and this is challenging to measure.

## 5. Conclusions

In summary, genetic variation in *TFIP11*, *MMP20*, and *ENAM* may have a protective effect increasing the chances of individuals preserving their teeth caries-free over a lifetime. These results show that these genomic biomarkers may be useful to determine individual susceptibility to dental caries, which may have consequences for predicting poorer overall health later in life.

## Figures and Tables

**Table 1 jpm-11-00023-t001:** Sex and ethnicity of individuals selected for the study by their phenotypic groups.

Phenotype	N	Sex		Ethnicity *		
		Female	Male	White	Black	Other
Edentulous by Age 30	30	12	18	30	0	0
Missing mandibular first molars by age 30	267	136	131	188	70	9
Missing mandibular second molars by age 30	180	98	82	130	45	5
Caries-free	161	87	74	127	31	3

* Note: White indicated self-reported European descent, Black indicated self-reported African American, and Other included self-reported Asians and American Indians.

**Table 2 jpm-11-00023-t002:** Genotyping distribution per tested phenotype.

Phenotype	SNP	Affected			Caries-Free at Age 60			*p*-Value
		AA *	AB	BB	AA	AB	BB	
Edentulous by age 30	rs3790506	4	11	9	6	25	35	0.35
	rs4694075	6	14	7	17	27	21	0.66
	rs12640848	7	7	16	22	28	25	0.15
	rs27565	5	11	5	15	23	25	0.35
	rs11362	6	12	8	19	27	22	0.83
	rs1784418	7	16	6	6	46	22	0.08
	rs3736309	2	12	16	4	15	53	0.12
	rs6574293	0	4	21	2	11	56	0.69
	rs10132091	5	10	9	13	36	20	0.65
	rs2619112	4	10	13	13	36	22	0.28
	rs7217186	3	11	9	16	37	15	0.23
	rs2235091	1	7	10	8	24	25	0.53
	rs198968	8	9	12	14	18	41	0.39
	rs5997096	0	5	16	2	14	47	0.71
Missing mandibular first molars by age 30	rs3790506	16	47	111	6	25	35	0.25
	rs4694075	42	89	48	17	27	21	0.52
	rs12640848	68	51	75	22	28	25	0.17
	rs27565	34	68	59	15	23	25	0.73
	rs11362	40	72	58	19	27	22	0.78
	rs1784418	17	118	60	6	46	22	0.97
	rs3736309	8	49	142	4	15	53	0.73
	rs6574293	4	34	142	2	11	56	0.83
	rs10132091	35	86	62	13	36	20	0.72
	rs2619112	36	105	49	13	36	22	0.7
	rs7217186	48	78	57	16	37	15	0.26
	rs2235091	14	51	59	8	24	25	0.83
	rs198968	45	31	111	14	18	41	0.29
	rs5997096	1	50	108	2	14	47	0.15
Missing mandibular second molars by age 30	rs3790506	12	35	81	6	25	35	0.31
	rs4694075	32	62	34	17	27	21	0.62
	rs12640848	54	34	55	22	28	25	0.1
	rs27565	31	50	39	15	23	25	0.62
	rs11362	35	54	35	19	27	22	0.82
	rs1784418	12	84	46	6	46	22	0.91
	rs3736309	7	35	105	4	15	53	0.87
	rs6574293	4	21	106	2	11	56	1
	rs10132091	29	62	42	13	36	20	0.75
	rs2619112	21	80	37	13	36	22	0.6
	rs7217186	33	60	38	16	37	15	0.46
	rs2235091	10	36	39	8	24	25	0.92
	rs198968	30	29	79	14	18	41	0.8
	rs5997096	0	39	76	2	14	47	0.03

* Note: AA and BB indicate each homozygotes state, and AB indicates the heterozygous state.

## Data Availability

The data presented in this study are available upon request.

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
