# Peer review of "Biomarkers for Lifetime Caries-Free Status"

_jpm, 2020, doi:10.3390/jpm11010023_

Round 1

Reviewer 1 Report

The topic is of interest as it deals with an important issue. The design of the study is fine, however the interpretation of the data (results) is weak. The analysis should be performed in depth to highlight most promising results. Moreover, the discussion should be extensively develop to give the reader new informations.

Author Response

The topic is of interest as it deals with an important issue. The design of the study is fine, however the interpretation of the data (results) is weak. The analysis should be performed in depth to highlight most promising results. Moreover, the discussion should be extensively develop to give the reader new informations.

RESPONSE: We added to the Discussion section on the interpretation of the results as requested.

Reviewer 2 Report

The authors aimed to test extreme outcomes of dental caries, such as edentulism at young age or prematurely losing permanent mandibular molars, to determine genetic biomarkers that can be used at the population level to identify individuals at higher risk for dental caries. The study is interesting, is easy to follow and covers an hot topic, but some issues should be improved before publication. Some typos should be corrected thorough the text. Discussion section: please briefly discuss some useful tools against dental caries, included the crosstalk with oral dysbiosis and role of SNP's in overall oral health (please see and discuss PMID: 31246083; PMID: 32560235; PMID: 32397555; PMID: 24843315) Conclusion Section is very short. Is necessary to improve it including some "take-home message".

Author Response

The authors aimed to test extreme outcomes of dental caries, such as edentulism at young age or prematurely losing permanent mandibular molars, to determine genetic biomarkers that can be used at the population level to identify individuals at higher risk for dental caries. The study is interesting, is easy to follow and covers an hot topic, but some issues should be improved before publication. Some typos should be corrected thorough the text. Discussion section: please briefly discuss some useful tools against dental caries, included the crosstalk with oral dysbiosis and role of SNP's in overall oral health (please see and discuss PMID: 31246083; PMID: 32560235; PMID: 32397555; PMID: 24843315) Conclusion Section is very short. Is necessary to improve it including some "take-home message".

RESPONSE: We revised the text and corrected all typos we could identify. We added to the discussion section using the references provided as requested. Similarly, we added to the conclusion statement as suggested.

Reviewer 3 Report

Thank you for the opportunity to review this manuscript.

The authors presented a study of the influence of genetic variation on extreme outcomes of dental caries.  

The topic is of great interest for better lifetime prevention of dental carries. This approach is an interesting addition to the current knowledge and published papers.

MS is generally very well written and easy to read, with no typos, errors or unclear information.

Authors well described their methodology and supported their discussion with obtained results. They acknowledge limitations of the study being local study on particular population.

There is one information, which is not clear –all data come from Dental Registry and DNA Repository project and author describe in their manuscript DNA analysis – do I understand it correctly that samples were analysed by authors for their aim? Or analysis was already performed and authors used data from the database? This should be made more clear for the reader.

The only issue in relation to the sample is unequal distribution of ethnical groups while concluding that ethnicity had no impact on the results. This limitation should be also discussed, as well as lower number of subjects in the Edentulous by age 30 group.

Similarly, authors also mention that all subjects in their study have low socio-economic status and hypothesised that their oral hygiene and diet are similar. As those two factors play very important role, which authors acknowledge it in their manuscript, it would be great to include them in the analysis. I understand that such data may not be available as authors sourced their data from the database, but it would be great addition to their manuscript.  

I suggest minor revision of the manuscript .

Author Response

Thank you for the opportunity to review this manuscript.

The authors presented a study of the influence of genetic variation on extreme outcomes of dental caries.  

The topic is of great interest for better lifetime prevention of dental carries. This approach is an interesting addition to the current knowledge and published papers.

MS is generally very well written and easy to read, with no typos, errors or unclear information.

Authors well described their methodology and supported their discussion with obtained results. They acknowledge limitations of the study being local study on particular population.

There is one information, which is not clear –all data come from Dental Registry and DNA Repository project and author describe in their manuscript DNA analysis – do I understand it correctly that samples were analysed by authors for their aim? Or analysis was already performed and authors used data from the database? This should be made more clear for the reader.

RESPONSE: We clarified what the reviewer requested. Data was analyzed for the purpose of the study. From the total participants on the registry at that time, 330 were selected for the study.

The only issue in relation to the sample is unequal distribution of ethnical groups while concluding that ethnicity had no impact on the results. This limitation should be also discussed, as well as lower number of subjects in the Edentulous by age 30 group.

RESPONSE: We added a comment on the Discussion as suggested.

Similarly, authors also mention that all subjects in their study have low socio-economic status and hypothesised that their oral hygiene and diet are similar. As those two factors play very important role, which authors acknowledge it in their manuscript, it would be great to include them in the analysis. I understand that such data may not be available as authors sourced their data from the database, but it would be great addition to their manuscript. 

RESPONSE: We added a comment in the Discussion section to address this concern. We did not have a direct measure of oral hygiene and diet to include in the analysis.

I suggest minor revision of the manuscript .

Reviewer 4 Report

Comment on manuscript:

This study addresses the hypothesis that individuals with genotypes of common variants in genes involved in weak dental enamel will present with extreme outcomes of dental caries. The study has been well-organized, but in my opinion, the following issues should be addressed:

  1. The abstract was too long. The abstract should be a total of about 200 words maximum
  2. The purpose sentence should be revised to be more understandable
  3. Please clearly state that the missing teeth due to other reasons than dental caries were excluded in this study
  4. The definition of ethnicity should be revised, the “white”, ”black” and ”other” may not suitable
  5. Line 86 and 88: Please provide manufacturing information for TaqMan chemistry and software PLINK
  6. Please explain why the p values below 0.003 (0.05/14) were considered significant in this study?
  7. Table 2: Please provide a footnote to explain the AA, AB, BB, etc.
  8. Please discuss further the limitations of the present studies and suggestions for further studies should be mentioned.
  9. Conclusion sentences should be revised to be more consistent with the purpose of this study

Author Response

Comment on manuscript (jpm-1033948-peer-review-v1):

This study addresses the hypothesis that individuals with genotypes of common variants in genes involved in weak dental enamel will present with extreme outcomes of dental caries. The study has been well-organized, but in my opinion, the following issues should be addressed:

1. The abstract was too long. The abstract should be a total of about 200 words maximum

RESPONSE: We shortened the abstract to 194 words.

2. The purpose sentence should be revised to be more understandable

RESPONSE: We revised the sentence as requested.

3. Please clearly state that the missing teeth due to other reasons than dental caries were excluded in this study

RESPONSE: We added the clarification as requested.

4. The definition of ethnicity should be revised, the “white”, ”black” and ”other” may not suitable

RESPONSE: We added a note under table 1 to clarify these definitions.

5. Line 86 and 88: Please provide manufacturing information for TaqMan chemistry and software PLINK

Response: We added the requested information.

6. Please explain why the p values below 0.003 (0.05/14) were considered significant in this study?

RESPONSE: We applied a correction for multiple testing. We added this clarification as requested.

7. Table 2: Please provide a footnote to explain the AA, AB, BB, etc.

RESPONSE: We added the explanation requested on Table 2.

8. Please discuss further the limitations of the present studies and suggestions for further studies should be mentioned.

RESPONSE: We added more discussion on the limitations of the study as suggested.

9. Conclusion sentences should be revised to be more consistent with the purpose of this study

RESPONSE: We revised the sentence as suggested.